# Drug Prescriptions among Italian and Immigrant Pregnant Women Resident in Italy: A Cross-Sectional Population-Based Study

**DOI:** 10.3390/ijerph19074186

**Published:** 2022-04-01

**Authors:** Paola D’Aloja, Roberto Da Cas, Valeria Belleudi, Filomena Fortinguerra, Francesca Romana Poggi, Serena Perna, Francesco Trotta, Serena Donati

**Affiliations:** 1National Centre for Disease Prevention and Health Promotion, Istituto Superiore di Sanità (National Institute of Health), 00161 Rome, Italy; serena.donati@iss.it; 2National Centre for Drug Research and Evaluation, Istituto Superiore di Sanità (National Institute of Health), 00161 Rome, Italy; roberto.dacas@iss.it; 3Department of Epidemiology, Lazio Regional Health Service, 00145 Rome, Italy; v.belleudi@deplazio.it (V.B.); f.poggi@deplazio.it (F.R.P.); 4HTA & Pharmaceutical Economy Division, Italian Medicines Agency (AIFA), 00187 Rome, Italy; f.fortinguerra@aifa.gov.it (F.F.); s.perna@aifa.gov.it (S.P.); f.trotta@aifa.gov.it (F.T.)

**Keywords:** pregnant women, drug prescription, prevalence, immigrants, pharmacoepidemiology, inappropriate prescribing

## Abstract

Ensuring drug safety for pregnant women through prescription drug monitoring is essential. The aim of this study was to describe the prescription pattern of medicines among pregnant immigrant women from countries with high migratory pressure (HMPCs) compared to pregnant Italian women. The prevalence of drug prescriptions among the two study populations was analysed through record linkage procedures applied to the administrative databases of eight Italian regions, from 2016 to 2018. The overall prevalence of drug prescription was calculated considering all women who received at least one prescription during the study period. Immigrants had a lower prevalence of drug prescriptions before (51.0% vs. 58.6%) and after pregnancy (55.1% vs. 60. 3%). Conversely, during pregnancy, they obtained a slightly higher number of prescriptions (74.9% vs. 72.8%). The most prescribed class of drugs was the blood and haematopoietic organs category (category ATC B) (56.4% vs. 45.9%, immigrants compared to Italians), followed by antimicrobials (31.3% vs. 33.7%). Most prescriptions were appropriate, while folic acid administration 3 months before conception was low for both study groups (3.9% immigrants and 6.2% Italians). Progesterone seemingly was prescribed against early pregnancy loss, more frequently among Italians (16.5% vs. 8.1% immigrants). Few inappropriate medications were prescribed among antihypertensives, statins and anti-inflammatory drugs in both study groups.

## 1. Introduction

In high-income countries, drug use during pregnancy is common; four out of five pregnant women receive at least one prescription [1]. Monitoring the use of each drug, including over-the-counter products and natural supplements, is strategic to determine their safety during pregnancy, especially during the first trimester, when the development of the embryo and foetus occurs. The impact of drug use on maternal and foetal health is a major concern, especially as pregnant women are excluded from clinical trials [2]. Considering pregnant women as a “vulnerable population protected by exclusion” [3] prevented research and the progression of care, even during the recent COVID-19 pandemic. Furthermore, the assessment of the prescription profile for subpopulations, such as migrants, provides useful information on their health status as compared to the host population, including their access to health services.

In 2020, approximately 5 million foreign citizens were registered in Italy, accounting for 8.4% of all residents; nearly half of these were women (51.7%), of whom more than 1.5 million were of child-bearing age [4]. Although migration has been recognized as contributing to the Sustainable Development Goals [5], migrants and refugees face barriers in accessing health services [6,7,8], and even in Europe they face obstacles to health care provision [9,10]. Immigrants from high migration pressure countries (HMPCs) often experience health care inequalities [11], including worse maternal and perinatal outcomes [9].

Although Italy offers universal access to health care including maternal care [12], differences in maternal and perinatal outcomes persist, according to women’s country of origin and/or citizenship [13,14]. Despite a decrease in the last 10 years, the average number of children (1.89) among foreign women remains higher compared to Italian mothers (1.17) [15], likely due to the younger age of migrant women and due to their different fertility model, which tends to anticipate the age at childbirth compared to Italian women. In 2020, mothers of foreign citizenship gave birth to 21% of Italian newborns [16]. 

In Italy, population-based studies on drug use during pregnancy are scanty, often linked to regional initiatives [17,18]. In 2018, the Italian Medicines Agency (AIFA) established the (Monitoring Medication Use During Pregnancy Network (MoM-Net), a national coordination group with the aim to monitor drug prescription patterns during pregnancy. MoM-Net activities included the analysis of: I) prescriptions in specific therapeutic areas; II) regional differences in prescriptions; III) prescriptions in particular subpopulations, such as foreign women and women with multiple pregnancies [19,20].

The primary aim of this paper is to describe the prescription pattern of medicines among pregnant immigrant women from HMPCs compared to Italian women; the secondary aim is to identify potentially inappropriate prescription areas among the two study populations.

## 2. Methods

Study Design: cross-sectional population-based study using the following regional administrative databases: Birth Registry (*Certificato di Assistenza al Parto*, *CeDAP*), including socio-demographic and health information of women who gave births ≥22 weeks of gestation and their newborns;Demographic database, including information on residents enrolled in the regional health system for administrative purposes;Drug prescription database, including prescriptions dispensed by pharmacies and reimbursed by the NHS, and describing date of issue, number of packages, active ingredients and brand.

Unique anonymised patient identifiers were used to link the databases at regional level adopting a common data model, as described in detail by a previous publication (20). 

Study setting and population: eight Italian regions, namely Lombardy, Veneto, Emilia-Romagna, Tuscany, Umbria, Lazio, Apulia and Sardinia participated in the study. Resident women aged 15 to 49 years, who gave birth to an alive or deceased infant from 1 April 2016 to 31 March 2018, were included in the study.

Resident women were identified by country of birth and citizenship and divided into two groups: Italians and immigrants from HMPCs (Central-Eastern Europe, Central and Southern America, North Africa, Sub-Saharan Africa and Asia (except for Israel and Japan)) [21]. For each woman, the date of onset of pregnancy was estimated according to the gestational age at time of delivery. The socio-demographic characteristics (e.g., age, nationality, education and occupational status), the clinical information related to pregnancy (e.g., gestational age and parity) and obstetric histories of the pregnant women (e.g., previous deliveries, previous caesarean sections, previous abortions retrieved from the Birth Registry database) were collected. Among multiparas, only the first birth was included in the analysis. Drug prescriptions were analysed adopting the anatomical therapeutic chemical (ATC) classification system [22] over a period of 27 months divided into three successive time windows: three trimesters before conception, three trimesters during pregnancy and three trimesters after birth. 

Statistical analysis: the overall and specific prevalence of medicines by target populations, namely Italians and immigrant women from HMPCs, was estimated as the percentage of women with at least one prescription during the study period (before, during and after pregnancy). According to the different therapeutic categories, the prevalence of drug use was stratified by maternal socio-demographic characteristics, reproductive history, and pregnancy information [20]. The χ^2^ test was used to evaluate statistical difference for continuous data: prevalence of use and percentages. Tests were carried out at 2-sided *p* < 0.05 level of significance. All statistical analyses were performed using R or SAS software (version 9.4).

## 3. Results

A total of 447,096 women who delivered were included in the study cohort, namely 358,467 Italian citizens, 2470 women, born abroad and in Italy, with citizenship in highly developed countries (HDCs), and 86,159 with HMPC citizenship. The present results describe the drug prescriptions issued to Italian and HMPC women together comprising 97.2% of resident pregnant migrant women enrolled in the cohort. The small number of HDC women (*n* = 2470) and their similar socio-demographic and health profiles with the enrolled Italian women were the rationale for their exclusion from the present analyses.

Overall, the births recorded in the participating regions accounted for 58.5% of total births that occurred in Italy during the study period. 

Table 1 shows the socio-demographic and obstetric characteristics of the two groups under consideration. Migrants from HMPCs were younger (aged < 35 years: 74.5% vs. 59.7%), less educated (≤8 years education: 43.4% vs. 19.4%) and more often multipara (66.8% vs. 45.5%) and unemployed (66.1% vs. 29.5%), compared to Italian women. Compared to Italian mothers, HMPC women rarely resorted to prenatal screening (5.8% vs. 13.3%) and assisted reproductive techniques (ART) (1.7% vs. 3.4%) and underwent fewer caesarean sections (27.7% vs. 31.0%).

Overall, women with HMPC citizenship received a lower number of prescriptions compared to Italians during the entire study period. Italian women registered higher prevalence of drug prescriptions before conception (58.6% vs. 51.0%) and after birth (60.3% vs. 55.1%), while HPMC women received the highest number of prescriptions during pregnancy (74.9% vs. 72.8%).

Figure 1 describes the distribution of drug prescriptions by trimester of analyses between the two populations. Overall, the highest prevalence of drug prescriptions occurred during pregnancy, with a peak in the first trimester (around 50%) and a sharp decline after the first postpartum trimester. Prevalence of drug use increased with increasing maternal age, reaching 60% in the women over 40 years of age (Appendix A). 

HMPC citizens had different prescriptive profiles depending on their geographical area of origin. Figure 2 illustrates the high variability recorded by citizenship among the top 25 countries by number of residents. Women from Africa and India, followed by South America, had the largest number of prescriptions. Eastern European women were in an intermediate position and Chinese women reported the lowest level of drug prescriptions. 

Overall, the most frequently prescribed class of drugs was the blood and haematopoietic organs category (ATC B), which includes folic acid, heparin and iron-based preparations. During pregnancy, HMPC women received the highest number of ATC B prescriptions compared to Italian women (56.4% vs. 45.4%) (Appendix A), largely for iron preparations (30.5% vs. 16.0%). The class of antimicrobials (ATC J) followed in frequency with a prevalence of about 30% among both populations (Appendix A). Compared to HMPC women, Italian women received a slightly higher prevalence of prescriptions, especially before conception (35.1% vs. 29.5%) (Figure 1). Antimicrobials were the most widely used drugs in the second trimester of pregnancy and in the first trimester post-partum (Appendix A). 

Table 2 shows the ranking of the 10 most prescribed substances during pregnancy between Italian and HMPC women. Folic acid was the most prescribed substance in both populations, followed in second position respectively by iron among HMPC women and progesterone among Italian women. In the rankings for HMPC women, amoxicillin plus clavulanic acid followed, which, together with amoxicillin and fosfomycin, described the principal classes of antimicrobials prescribed to this population. Levothyroxine, in fifth position among Italian prescriptions, ranked seventh among prescriptions to HMPC women. Low molecular weight heparin ranked tenth among Italian women and twelfth among HMPC women. Progesterone ranked second among Italian women and fourth among HMPC women.

HMPC women received a higher proportion of anti-inflammatory drug prescriptions across all study periods, compared to Italian women (6.5% vs. 5.5% in pre-pregnancy, 2.1% vs. 1.2% in pregnancy and 4.2% vs. 3.8% in post-partum). Overall, the use of psychotropic drugs decreased during pregnancy and was slightly lower among HMPC compared to Italian women (0.9% vs. 1.2% during pregnancy; 1.5% vs. 1.7% post-partum). Selective serotonin reuptake inhibitors (SSRIs) were the most commonly prescribed drugs among antidepressants during the three study periods.

During all study periods, HMPC women received a higher proportion of antidiabetic prescriptions than Italian women, with a maximum increase during pregnancy (4.4% vs. 2.2%).

Italian women received a higher proportion of heparin prescriptions than HMPC women during all study periods (2.1% vs. 1.4% before pregnancy, 5.7% vs. 2.8% during pregnancy and 23.5% vs. 19.2% postpartum). Prescriptions for low molecular weight heparin (LMWH) after CS were provided to 60% of Italian women and 54% of HMPC women, with similar age distribution.

The prescriptions of thyroid preparations in pregnancy doubled compared to their numbers during the pre-conception period, with higher percentages among Italian women compared to HMPC women, especially in the second trimester of pregnancy (4.9% vs. 2.1% pre-pregnancy and 8.3% vs. 5.4% during pregnancy). 

During pregnancy, HMPC women received more prescriptions for antacid drugs than Italian women (7.8% vs. 5.3%) and had a slightly higher prevalence of antihypertensive prescriptions (2.08% vs. 1.93%). 

As for inappropriate prescriptions, no alarming data were detected among the two study populations. Almost none of the antibiotic prescriptions during pregnancy showed inappropriateness in terms of teratogenic risk molecules. Among women affected by hypertensive disorders of pregnancy, a small proportion of ACE inhibitors and angiotensin II receptor blocker prescriptions were detected during pregnancy, and were slightly higher among HMPC women (2.08%) than Italian women (1.93%). The inappropriate prescription of progestogens during the first trimester of pregnancy was high, especially among Italian women (17.1%) compared to HMPC women (8.2%).

## 4. Discussion

This article focused on medical prescriptions among foreign pregnant women in eight regions of Italy. To our knowledge, this is the first study on the prevalence of drug prescriptions within such a large population of HMPC women in Italy; the previous studies were in fact limited to a single region [17,18]. The prescription pattern of medicines during pregnancy among HMPC women, compared to that of Italian women, showed some differences. HMPC women received more prescriptions for iron preparations, drugs to counteract acid secretion disorders, and anti-inflammatory, antihypertensive and antidiabetic drugs. Italian women received the greatest number of prescriptions for progesterone, antimicrobials, thyroid preparations, heparin and psychotropic drugs. Potentially inappropriate prescription areas were detected in a minority of cases. 

The major strength of our study is the population size, covering 58.5% of births in Italy during the study period and the participation of regions from all geographical areas of the country, avoiding possible bias due to different prescribing patterns between the northern and southern regions. The study limitations include the exclusion of foreign non-resident women who were not registered in the available administrative databases. However, Italian law provides access to free assistance during pregnancy, childbirth and up to the age of six months of the newborn for all women, including foreigners without a residence permit. It seems, therefore, unlikely that this exclusion could be a potential source of bias. The lack of information about over-the-counter drugs and about therapeutic indications for the prescribed drugs did not allow an accurate assessment of their clinical appropriateness. Moreover, the same databases did not provide information about prescriptions in case of miscarriage or induced abortion, preventing the assessment of drugs’ possible teratogenic risk. 

In previous studies, drug use in pregnancy ranged from 50% to 80% depending on the setting; when over-the-counter treatments were included, rates reached 100% [1,9,18,23,24]. In general, the overall rate and prescription patterns observed in this study were comparable to those observed in other European studies [25,26].

The detected prescriptive pattern for the HMPC pregnant women seemed to outline different health profiles and different opportunities to access care during pregnancy compared to Italian women. Our analysis showed that among HMPC women, the use of drugs was lower during pregnancy due to an increase in iron and folic acid prescriptions. This was probably due to the high prevalence of iron-deficiency anaemia among migrants, particularly critical for women from Africa and Southeast Asia, due to a diet poor in foods containing this mineral [27].

The low prescription of folic acid during the pre-conception period affected both populations. The Italian National Health Service offers this vitamin free of charge in order to promote neural tube defect prevention. Nevertheless, the consumption of folic acid in the pre-conception period was, overall, very low and lower than in pregnancy, especially among HMPC women (3.9% vs. 6.2% among Italian women). Previous Italian sample surveys carried out by interviewing women who gave birth [13] detected a 20% prevalence of appropriate use of folic acid during the peri-conceptional period. The lowest prevalence detected by the present analysis was likely attributable to the over-the-counter availability of folic acid, which is often purchased out-of-pocket, especially among Italian women. This result suggests the need to better inform Italian and foreign women of childbearing age about the advantage of taking this vitamin to prevent congenital defects.

Although studies have shown that administration of progesterone during the first trimester of pregnancy does not reduce the incidence of miscarriages, except for recurrent cases [28,29], our analysis showed that it is still prescribed to prevent this outcome [30]. Prescriptions are though to be more common among women who have had recurrent abortions in line with evidence-based recommendations. In addition, the more frequent use of progesterone among Italian women is probably associated with the greater number of assisted human reproduction procedures that these women underwent.

About one in three women received an antibiotic in both populations, with a slightly higher prevalence among Italians, likely due to their most frequent prescription as prophylaxis in case of invasive prenatal diagnosis or caesarean section, which are less common among HMPC women. However, it should be pointed out that a significant share of antimicrobials are purchased privately in Italy. Therefore, the differences observed between the Italian and HMPC women could be wider because the latter, due to their lower income status, rely more on the prescriptions charged to the NHS. The detected 30% prescription rate was consistent with the use of antimicrobials in Europe, ranging from 27% to 42% [31]. Since most of these prescriptions reached the first trimester of pregnancy, it seems possible that they were “prescriptive queues” of previous therapies that had been stopped following the diagnosis of pregnancy. Overall, the antibiotics prescribed in pregnancy were appropriate, although it is necessary to sensitize health professionals to the urgency of limiting prescriptions during pregnancy and to carefully choose the molecules to be used to limit serious consequences related to the growing phenomenon of antibiotic resistance.

Although previous studies have associated belonging to minority ethnic groups, lower socio-economic status and lack of language skills with a higher risk of perinatal mental disorders [32], the present analysis found slightly lower prescriptions of psychotropic drugs among HMPC women compared to Italian women. The difficulties in accessing health services likely play a role in this difference, which could be primarily attributable to unmet needs of the HMPC women [33,34]. The wide variability in prescription drug profiles by women’s geographic origin suggests different challenges in accessing obstetric care. 

Overall, the areas of inappropriate treatment during pregnancy did not show significant differences according to the women’s countries of birth. Nevertheless, the few detected prescriptions related to drugs with a critical safety profile in pregnancy—including statins, ACE inhibitors, angiotensin II receptor antagonists, and anti-inflammatories—were slightly higher among foreign women than Italians. However, almost all of these prescriptions concerned the first trimester of pregnancy, and it is, therefore, reasonable to imagine that they were “prescriptive queues” of previous therapies that had been changed following the diagnosis of pregnancy. The antihypertensive drugs that can be used safely in the first trimester of pregnancy include methyldopa, labetalol and nifedipine. ACE inhibitors and angiotensin II receptor antagonists [35,36], on the other hand, should be suspended when planning or establishing a pregnancy due to association with a higher incidence of cardiovascular and central nervous system congenital malformations [37]. During pregnancy, and in particular in the third trimester, the consumption of anti-inflammatory drugs is considerably reduced among all women, due to a greater risk of adverse events at birth. Anti-inflammatory prescriptions decline in pregnancy due to their critical safety profile but are still prescribed to 2% of HMPC women. 

Among the challenges for measuring Europe’s reception capacity and its commitment to integration and social development is that of guaranteeing full equity of access to care services for all women and their children, without differences based on origin or social status, and with equal dignity and security. Pregnancy and childbirth represent a period of vulnerability for migrant women experiencing communication and language barriers that may create disadvantages in establishing relationships with their health care providers [9,38,39,40,41]. Migrant women have worse maternal and perinatal outcomes compared to Italian women for reasons attributable to lower income levels, problems in accessing and using some care opportunities and greater precariousness of the family support network [9,34,42]. The “healthy migrant effect” self-selects at the origin the people in good health that face the migration path; however, once in the host countries, the health of migrants tends to worsen due to precarious living conditions, social exclusion and fragility, and the acquisition of unhealthy lifestyles (e.g., eating habits, sedentary lifestyle) [43].

This analysis, as part of a larger study conducted by the MoM-Net group, described the prescribing pattern of medications during pregnancy in sub-groups of the population by identifying potentially inadequate prescriptions. Characterizing the health profile of foreign pregnant women, including monitoring of drug prescriptions, is strategic not only to improve maternal and perinatal care and outcomes but also to support policies for the provision of social and health services in Italy, where the migratory phenomenon represents a structural element of society, with over 5 million resident foreign citizens.

## Figures and Tables

**Figure 1 ijerph-19-04186-f001:**
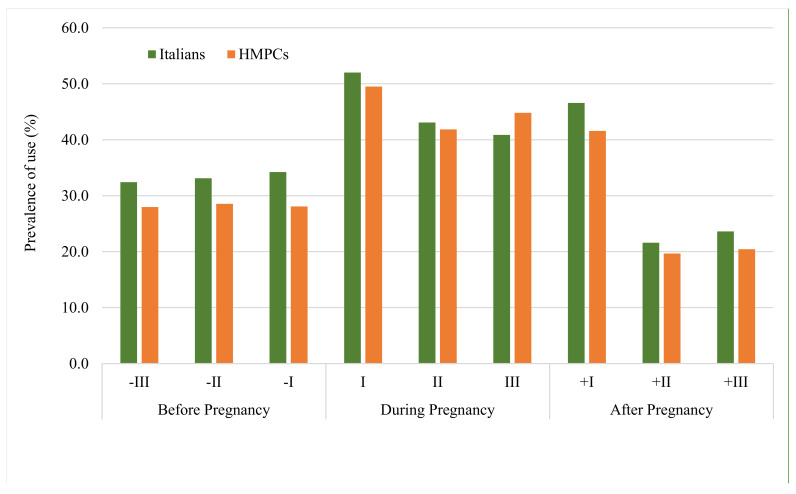
Prevalence of drug use (%) by trimester before, during and after pregnancy.

**Figure 2 ijerph-19-04186-f002:**
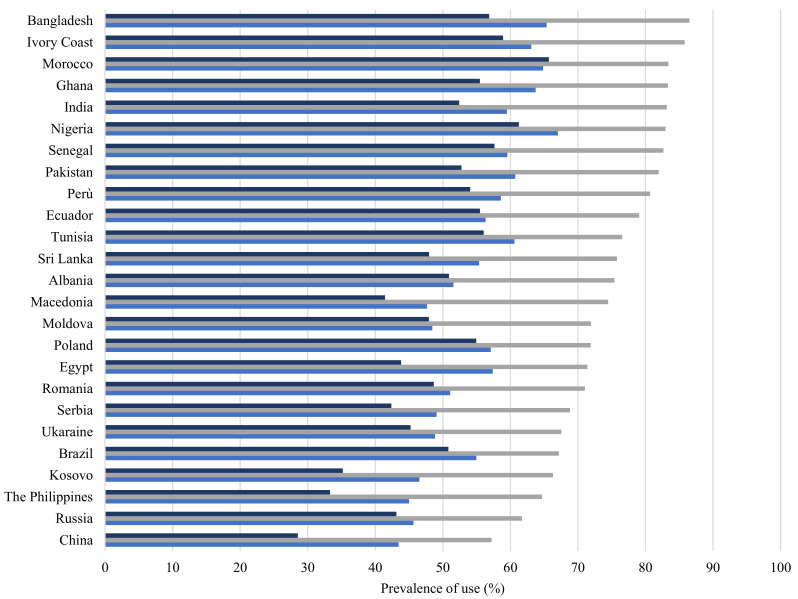
Distribution of drug prescriptions (%) before, during and after pregnancy, by citizenship (top 25 countries by number of residents).

**Table 1 ijerph-19-04186-t001:** Socio-demographic characteristics and obstetric history of Italian and HMPC women.

Characteristics				
	Italians		HMPCs	
	N = 358,467		N = 86,159	
	*n*	%	*n*	%
**Age group**				
≤24	21,911	6.1	11,512	13.4
25–29	66,117	18.4	25,524	29.6
30–34	126,028	35.2	27,154	31.5
35–39	105,977	29.6	17,230	20.0
≥40	38,434	10.7	4739	5.5
≥45	3100	8.1	288	6.1
**Level of education**				
None/elementary school/ Middle school	68,063	19.4	36,757	43.4
High school	160,285	45.8	34,551	40.8
Bachelor degree /post degree	121,493	34.7	13,115	15.5
Missing	479	0.1	193	0.2
**Occupational status**				
Employed	252,765	70.5	29,183	33.9
Unemployed/ Looking for first job	42,098	11.7	11,993	13.9
Housewife	54,731	15.3	42,999	49.9
Other	5898	1.6	1233	1.4
Missing	2975	0.8	751	0.9
**Previous delivery**				
no	190,759	54.5	28,096	33.2
yes	159,561	45.5	56,520	66.8
Caesarean section	44,429	27.8	14,291	25.3
**Previous abortions**				
0	290,138	80.9	66,844	77.6
1	51,317	14.3	14,133	16.4
2+	17,012	4.7	5182	6.0
**Gestational age**				
Preterm delivery (<37 weeks)	23,976	6.7	6497	7.5
Term delivery (37–41 weeks)	332,267	92.7	79,043	91.7
Post-term delivery (>41 weeks)	2224	0.6	619	0.7
**Parity**				
1	351,687	98.1	84,772	98.4
2+	6780	1.9	1387	1.6
**Invasive antenatal diagnosis**				
No	309,995	86.5	80,833	93.8
Chorionic villus sampling	18,650	5.2	1626	1.9
Amniocentesis	28,091	7.8	3064	3.6
Other	1137	0.3	289	0.3
**Medically assisted procreation**				
no/n.c.	281,903	96.6	75,026	98.3
yes	9787	3.4	1315	1.7
**Caesarean section**				
no	247,309	69.0	62,271	72.3
yes	111,158	31.0	23,888	27.7

**Table 2 ijerph-19-04186-t002:** Ranking of the most 10 prescribed substances during pregnancy by citizenship.

Rank	Substances	Italians N = 358,467			HMPC N = 86,159			
		*n*	Prevalence of Use (%)	Rank	*n*	Prevalence of Use (%)	Rank	*p* Value ***
1	Folic acid	119,035	33.2	1	34,906	40.5	1	<0.05
2	Progesterone	74,452	20.8	2	9919	11.5	4	<0.05
3	Ferrous sulfate	57,190	16.0	3	26,253	30.5	2	<0.05
4	Amoxicillin/clavulanic acid	40,760	11.4	4	10,256	11.9	3	<0.05
5	Levothyroxine	29,535	8.2	5	4604	5.3	7	<0.05
6	Fosfomycin	28,214	7.9	7	6358	7.4	6	<0.05
7	Azithromycin	25,626	7.1	6	3750	4.4	8	<0.05
8	Amoxicillin	22,899	6.4	8	6376	7.4	5	<0.05
9	Beclometasone	17,768	5.0	9	2806	3.3	9	<0.05
10	Enoxaparin	15,990	4.5	10	1982	2.3	12	<0.05

* *p* value from χ^2^ test.

## Data Availability

The data that support the findings of this study are available from the Italian AIFA Medicine Agency and are available from the authors upon reasonable request and with permission of the AIFA and the participating Regions; Requests to access the datasets should be directed to v.belleudi@deplazio.it.

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
