# Peer review of "Drug Prescriptions among Italian and Immigrant Pregnant Women Resident in Italy: A Cross-Sectional Population-Based Study"

_ijerph, 2022, doi:10.3390/ijerph19074186_

Round 1
Reviewer 1 Report
Dear Author
Thank you for the opportunity to review this very interesting manuscript.
I think it is very significant in the field of public health to identify drug prescriptions among immigrant pregnant women.
As a result of my review, I would like to make the following suggestions.
<Major>
- I think it is necessary to show statistically (e.g., chi-square test) the difference of the surveyed population shown in Table 1. In that case, I think you should describe the statistical method in the Method section.
- This study compares the differences in drug prescriptions between Italians and HMPCs, but I think it needs to be compared using statistical methods (e.g., cross tabulation tables and chi-square test as in Table 1).
- It would be necessary to indicate pregnancy complications and other medical history because the objective of this study is drug prescriptions.
<Minor>
- What is the difference between “previous delivery” and “parity” shown in Table 1? In addition, is there any population with parity of “0” in this study?
- Are there any exclusion criteria for this study? If yes, please indicate the exclusion criteria for the study subject.
- The title of figures should be shown below the figure.
- It is better to indicate what the horizontal axis in Figure 2 represents.
- L152: Where is the data (table or figure) for “largely for iron preparations”?
- Where is the data (table or figure) showing the contents of L154-155?
- Where is the data (table or figure) showing the contents of L169-195?
- HMPC and HMCP are mixed.
Author Response
Reply: We thank the reviewers for their careful reading of the manuscript and their constructive remarks. We went through the comments to improve and clarify the manuscript. Please find below a detailed point-by-point response to all comments (reviewers’ comments in black, our replies in blue). Line numbering refers to the revised manuscript.
Review #1
We would like to thank the reviewer #1 for his/her valuable comments on the manuscript. In response to these comments, we have amended the text as follows.
<Major>
I think it is necessary to show statistically (e.g., chi-square test) the difference of the surveyed population shown in Table 1. In that case, I think you should describe the statistical method in the Method section.
Response: Thanks the Reviewer # 1 for this comment. For all variables in Table 1, the statistical difference expressed as a P-value is added (calculated using the chi-square test) (page n.5). The Methods section was modified with the addition of the statistical analysis paragraph (lines 145, 150-153).
This study compares the differences in drug prescriptions between Italians and HMPCs, but I think it needs to be compared using statistical methods (e.g., cross tabulation tables and chi-square test as in Table 1).
Response: We thank the Reviewer # 1 for raising this critical issue. Statistical significance was not included in the results of the submitted version, as the study did not examine a sample of women. All the women, Italians and migrants, living in 8 regions aged 15 and 49 years, who had a child during the period of study, were considered in the analysis. After this suggestion, in the revised manuscript we have estimated statistical difference as showed in Tables 1 and 2 (pages 5 and 8)
It would be necessary to indicate pregnancy complications and other medical history because the objective of this study is drug prescriptions.
Response: We thank Reviewer # 1 for raising that point. The aim of the study was to describe the prescriptive pattern of immigrant women compared to Italians, so it didn't include outcomes related to drug prescriptions. In addition, discharge data were not gathered in the present study. The regional birth register can only inform on the presence of previous deliveries, number of previous abortions (spontaneous and not), gestational age at the time of childbirth, number of births, invasive prenatal examinations, medically assisted reproduction and type of childbirth (caesarean and not).
To clarify this point, in the methods section, we have more clearly defined the information available from the databases used in the study (lines 136-140).
<Minor>
What is the difference between “previous delivery” and “parity” shown in Table 1? In addition, is there any population with parity of “0” in this study?
Response: We thank the reviewer # 1 for this question. In table 1, under the heading “previous delivery”, a distinction is made between women who had not given birth before the study period and who had given birth in the past, with or without a C-section. Moreover, regarding the question on women with parity of “0”, as mentioned in the Methods, the study included only “All resident women aged 15 to 49 years, who gave birth to an alive or deceased infant from April 1, 2016 to March 31, 2018,”. Therefore, nulliparous women are excluded from the study.
Are there any exclusion criteria for this study? If yes, please indicate the exclusion criteria for the study subject.
Response: Thank you for pointing this out. In our study, data on induced abortions and miscarriages were not available. To make it clearer, the following sentence has now been included in the Methods: “Voluntary abortions and miscarriages were not included, as this information were not recorded in the Birth Registry database” (lines 137-138)
The title of figures should be shown below the figure.
Response: Thank you for this suggestion. We proceeded to move the title of figures below the figure.
It is better to indicate what the horizontal axis in Figure 2 represents.
Response: Thank you for this suggestion. As the reviewer commented, we modified the figure and legend specifying what the horizontal axis represents “Prevalence of use (%)” (page 7)
L152: Where is the data (table or figure) for “largely for iron preparations”?
Where is the data (table or figure) showing the contents of L154-155?
Where is the data (table or figure) showing the contents of L169-195?
HMPC and HMCP are mixed.
Response: We apologize for our carelessness and thank the reviewer # 1 for pointing this out. In this new version, the missing data are reported and the figures in which the data are presented are provided (lines 215-219)
Reviewer 2 Report
Dear Authors,
Dear Authors,
The article is very well organized, the presentation of the inclusion-exclusion criteria well informed, as well as the comprehensive bibliography for this extensive study.
From my point of view, some small clarifications / adjustments are needed:
Line 146 - figure 2, in order to have a better visibility it is recommended to color differently, so that the periods before pregnancy and after pregnancy they are more delimited
Line 167 - The authors should explain the correlation between the representation in Figure 3 (the most prescribed drugs during pregnancy) in Italian women versus HMPC women, and the representation in Figure S2. Prevalence of antibiotic prescription by trimester before, during and after pregnancy. Is there a correlation? Given that in figure S2 Italian women predominate in the category of prescribed antibiotics (before pregnancy, during pregnancy and after birth) than HMPC.
Line 227 - The authors refer to a previous study on HMPC (according to bibliographic index 28, from 2012): it would be useful to know if other studies have been done so far to which reference can be made in this article.
Line 283-289 - The authors refer to the antihypertensives indicated in pregnancy and especially in the first trimester of pregnancy. However, the prevalence of hypertension in connection with critical conditions such as preeclampsia, a thrombotic disorder associated with hypertension that are usually present in the second half of pregnancy, may account for a significant percentage of indications. Can authors present data on drugs for these risk categories? [bibliographic index 36-37-38].
Your sincerely,
Adriana Chis
Author Response
Reply: We thank the reviewers for their careful reading of the manuscript and their constructive remarks. We went through the comments to improve and clarify the manuscript. Please find below a detailed point-by-point response to all comments (reviewers’ comments in black, our replies in blue). Line numbering refers to the revised manuscript.
Review #2
We would like to thank reviewer #2 for her valuable feedback on the manuscript. To respond to the concerns raised, the following changes have been added to the text.
Line 146 - figure 2, in order to have a better visibility it is recommended to colour differently, so that the periods before pregnancy and after pregnancy they are more delimited
Response: We appreciate your suggestion. The colour of the bar "before pregnancy" has changed to make the distinction between "before pregnancy" and "after pregnancy" clearer (page 7).
Line 167 - The authors should explain the correlation between the representation in Figure 3 (the most prescribed drugs during pregnancy) in Italian women versus HMPC women, and the representation in Figure S2. Prevalence of antibiotic prescription by trimester before, during and after pregnancy. Is there a correlation? Given that in figure S2 Italian women predominate in the category of prescribed antibiotics (before pregnancy, during pregnancy and after birth) than HMPC.
Response: Thank the Reviewer #2 for raising this issue. Figure 3 provided a comparison of the most commonly prescribed substances between the two populations during pregnancy (ranking in terms of prevalence of use). Figure S2 deals with the prevalence in percentage of the antibiotic category prescribed between the two populations during, before and after pregnancy, by trimesters. Figures represent different, but related observations. The antibiotic category is the second largest during pregnancy (see Figure S1). Among the most frequently prescribed molecules (Figure 3), four antibiotics are included, confirming that the class of such drugs is widely prescribed. To better illustrate the most prescribed substances during pregnancy among the two study populations, we replaced Figure 3 with a table (Table 2) including information about the number of women who received at least one prescription and prevalence of use.
Line 227 - The authors refer to a previous study on HMPC (according to bibliographic index 28, from 2012): it would be useful to know if other studies have been done so far to which reference can be made in this article.
Response: Thanks the Reviewer #2 for reporting the issue. The citation was in the wrong position. This error has now been corrected: reference 28 is now 44. On including additional studies in the reference, to our knowledge, there are no other studies available on the prescription pattern before, during and after pregnancy in immigrant pregnant women compared to the native population. Some are limited to the period of pregnancy; others are specific to certain categories of drugs.
Line 283-289 - The authors refer to the antihypertensives indicated in pregnancy and especially in the first trimester of pregnancy. However, the prevalence of hypertension in connection with critical conditions such as preeclampsia, a thrombotic disorder associated with hypertension that are usually present in the second half of pregnancy, may account for a significant percentage of indications. Can authors present data on drugs for these risk categories? [bibliographic index 36-37-38].
Response: Thanks for pointing out this point. As we answer to the Reviewer 1 The aim of the study was to describe the prescriptive pattern of immigrant women compared to Italians, so it didn't include outcomes related to drug prescriptions. In addition, discharge data were not gathered in the present study. The regional birth register can only inform on the presence of the previous deliveries, number of previous abortions (spontaneous and not), gestational age at the time of childbirth, number of births, invasive prenatal examinations, medically assisted reproduction and type of childbirth (caesarean and not) (lines 138-140).
Reviewer 3 Report
It is a very interesting topic for potential readers of this publication. Some minor comments are made in order to improve the current version of the manuscript:
. Title. Perhaps it's better “italian” than “Italian”.
.- Keywords. It could be removed “Italy”; “drug prescription” instead of “prescription medicines”; “inmigrant pregnants” instead of “Inmigrants”; “inappropiate prescription” instead of “inappropiate prescribing”
.- Abstract. Important doubt, what are the objectives of this study?. The prevalence of drug prescription? know the most prescribed drugs in one subpopulation and in another one?
When folic acid and progesterone levels are mentioned, perhaps percentages could be added. In the results, it is missing if statistical significance was obtained.
.- Introduction. It's so large. Perhaps the first three paragraphs could be summarized/shortened.
.- Methodology. why those dates were chosen and not others? It is missing to specify statistical analysis and statistical package used.
.- Results. It is missing if statistical significance was obtained or not in the exposed results (p equal to or less than 0.5)
.- Discussion. It's so large.
.- Conclusion. Unclear conclusions since the objectives are not well specified.
. Figure 3. Perhaps it would add more value to the manuscript to specify absolute values, percentages and significant differences (if they were obtained) than only the ranking of the most prescribed drugs.
.- Bibliography. Of the 43 citations, 18 are recent (less than 5 years). Perhaps it could be valued to include some additional recent citation. In some references dates appear, in other ones do not appear. Review all references as well as author publication guide.
Author Response
Reply: We thank the reviewers for their careful reading of the manuscript and their constructive remarks. We went through the comments to improve and clarify the manuscript. Please find below a detailed point-by-point response to all comments (reviewers’ comments in black, our replies in blue). Line numbering refers to the revised manuscript.
Review #3
We would like to thank reviewer #1 for her/his valued input on the manuscript. In response to these comments, the following changes were made to the text.
Title. Perhaps it's better “italian” than “Italian”.
Response: Thanks the reviewer #3 for the suggestion. We changed the title, as was pointed out.
Keywords. It could be removed “Italy”; “drug prescription” instead of “prescription medicines”; “inmigrant pregnants” instead of “Inmigrants”; “inappropiate prescription” instead of “inappropriate prescribing”
Response: Thank the Reviewer #3 for the valuable input from the reviewer. “Italy” has been removed. We have replaced "prescription medicines" with "drug prescriptions". About the rest, we have left other keywords unchanged as they are MeSH Terms.
Abstract. Important doubt, what are the objectives of this study? The prevalence of drug prescription? know the most prescribed drugs in one subpopulation and in another one? When folic acid and progesterone levels are mentioned, perhaps percentages could be added. In the results, it is missing if statistical significance was obtained.
Response: Thank you for pointing out these critical issues. In response to the point about the lack of the object of this study in the abstract, the next sentence has been included in the paragraph “The aim of this study is to describe the prescription pattern of medicines among pregnant immigrant women from countries with high migratory pressure (HMPCs) compared to pregnant Italian women” (lines 18-22). Also, the percentages of folic acid and progesterone are now added to the abstract section (line 29). Statistical significance was not included in the results of the submitted version, as the study did not examine a sample of women. Actually, all women, Italians and migrants, living in 8 regions aged 15 and 49 years, who had a child during the period of study, were considered in the analysis. In the revised manuscript we have estimated statistical difference as showed in Tables 1 and 2 (lines 138-140).
Introduction. It's so large. Perhaps the first three paragraphs could be summarized/shortened.
Response: Thanks the reviewer for his suggestion. We carefully reviewed the introduction and cut out two sentences to make the section more concise (line 74 and 83).
Methodology. why those dates were chosen and not others? It is missing to specify statistical analysis and statistical package used.
Response: Thanks for pointing out this point. As we have told other the aim of the study was to describe the prescriptive pattern of immigrant women compared to Italians, so it didn't include outcomes related to drug prescriptions. In addition, discharge data were not gathered in the present study. The regional birth register can only inform on the presence of the previous delivery, number of previous abortions (spontaneous and not), gestational age at the time of childbirth, number of births, invasive prenatal examinations, medically assisted reproduction and type of childbirth (caesarean and not) (lines 138-140). To clarify statistical analysis, in the methods section, we have more clearly defined the information available from the databases used in the study.
To clarify statistical analysis performed, we modified the Methods section with the addition of the paragraph on statistical analysis (lines 145, 150-153).
Results. It is missing if statistical significance was obtained or not in the exposed results (p equal to or less than 0.5)
Response: Statistical significance was not included in the results, as the study did not examine a sample of women. Here all the women, Italians and migrants, living in 8 regions aged 15 and 49 years, who had a child during the period of study, were considered. In the revised manuscript we have estimated statistical difference expressed as P value (calculated through chi-square test) as showed in Tables 1 and 2 and modified the Methods section with the addition of the paragraph on statistical analysis (pages 5 and 8).
Discussion. It's so large.
Response: the reviewer's comment is greatly appreciated. This discussion was carefully reviewed. We reshaped it by removing a few redundant sentences (lines 359, 416, 420). In our opinion, making it too synthetic may remove important considerations on the appropriateness of drug use during pregnancy. We believe that an in-depth discussion can also be helpful in updating professionals in the field.
Conclusion. Unclear conclusions since the objectives are not well specified.
Response: In response to the point about the lack of the object of this study in the abstract, the next sentence has been included in the paragraph “The aim of this study is to describe the prescription pattern of medicines among pregnant immigrant women from countries with high migratory pressure (HMPCs) compared to pregnant Italian women” (lines 18-22).
Figure 3. Perhaps it would add more value to the manuscript to specify absolute values, percentages and significant differences (if they were obtained) than only the ranking of the most prescribed drugs.
Response: Thanks the reviewer #3 for this suggestion. To better illustrate the most prescribed substances during pregnancy among the two study populations, Figure 3 has been replaced by a table (Table 2) (page 8) with information on the number of women with at least one prescription, prevalence of use and P-value.
Bibliography. Of the 43 citations, 18 are recent (less than 5 years). Perhaps it could be valued to include some additional recent citation. In some references dates appear, in other ones do not appear. Review all references as well as author publication guide.
Response: Thanks the reviewer # 3 for raising this point. We have reviewed all references in accordance with the author’s guidance. In addition, there are several updated references: 6, 23, 25, 38, 40.
Round 2
Reviewer 1 Report
Thank you very much for your correction. I have read the revised draft and would like to comment as follows.
・Thank you for the correction of Table 1. However, since the chi-square test cannot test between each group, we do not believe it is appropriate to have p-values displayed for all groups.
・Thank you for correcting Table 2. I think it became clear that there is a bias in drug prescriptions for Italians and HMPC by your correction.
・Thank you for correcting the position of figure title. However, table title is fine as it was before the correction.
・Thank you for the other minor corrections.
Author Response
Thank you for the second round of Reviewer#1 comments and the opportunity to further revise the manuscript. Structure the manuscript in a manner that is most pertinent and understood by readers is a high priority for my co-authors and myself. We appreciate the time and effort that Reviewer #1 has invested in this task. Please find below a detailed point-by-point response to all comments (reviewers’ comments in black, our replies in blue).
Thank you for the correction of Table 1. However, since the chi-square test cannot test between each group, we do not believe it is appropriate to have p-values displayed for all groups.
Response: Thanks to Review # 1 for her/his help. As a result of the suggestion, the p-value of Table 1 was deleted in the revised manuscript.
Thank you for correcting Table 2. I think it became clear that there is a bias in drug prescriptions for Italians and HMPC by your correction.
Response: Thank you for your input in that regard. We also think that the p-value column should be left in Table 2. The differences in the prescription pattern for the specific substances (e.g. folic acid) between Italians and HMPC are commented in the discussion section. However, we are available for further clarification on the analysis.
Thank you for correcting the position of figure title. However, table title is fine as it was before the correction.
Response: We apologize for having interpreted that, also with respect to the tables, the titles should appear below the table. In the revised manuscript, they appear as in the original manuscript.